# Dosimetric and Clinical Prognostic Factors in Single-Isocenter Linac-Based Stereotactic Radiotherapy for Brain Metastases

**DOI:** 10.3390/cancers16183243

**Published:** 2024-09-23

**Authors:** Valeria Faccenda, Riccardo Ray Colciago, Sofia Paola Bianchi, Elena De Ponti, Denis Panizza, Stefano Arcangeli

**Affiliations:** 1Medical Physics Department, Fondazione IRCCS San Gerardo dei Tintori, 20900 Monza, Italy; valeria.faccenda@irccs-sangerardo.it (V.F.); elena.deponti@irccs-sangerardo.it (E.D.P.); 2School of Medicine and Surgery, University of Milan Bicocca, 20126 Milan, Italy; riccardo.colciago@unimib.it (R.R.C.); stefano.arcangeli@unimib.it (S.A.); 3Radiation Oncology Department, MedAustron Ion Therapy Center, 2700 Wiener Neustadt, Austria; sofia-paola.bianchi@medaustron.at; 4Radiation Oncology Department, Fondazione IRCCS San Gerardo dei Tintori, 20900 Monza, Italy

**Keywords:** Linac-based, stereotactic radiotherapy, brain metastases, treatment outcomes, prognostic factors

## Abstract

**Simple Summary:**

Although some of the novel systemic treatments, especially the group of tyrosine kinase inhibitors, have shown a durable central nervous system response, ionizing radiation remains the mainstay in the management of brain metastases (BM). Recent technological advancements have enabled the replacement of whole-brain radiotherapy with localized stereotactic radiotherapy (SRT) for treating up to 10 BM, either as a primary or combined treatment, reducing neurotoxicity and improving local control (LC). The delivered target dose and patient selection play a crucial role in enhancing treatment efficacy. However, there is still limited evidence supporting which factors most affect LC and which patients derive the greatest benefit from SRT. This retrospective single-institutional study evaluated treatment outcomes in a heterogeneous patient population treated with Linac-based SRT, with the aim of identifying potential dosimetric and clinical prognostic factors to better inform the decision-making process.

**Abstract:**

**Background/Objectives:** To report on predictive factors in Linac-based SRT for single and multiple BM. **Methods:** Consecutive patients receiving either one or three fractions of single-isocenter coplanar VMAT SRT were retrospectively included. The GTV-PTV margin was 1–2 mm. The delivered target dose was estimated by recalculating the original plans on roto-translated CT according to errors recorded by post-treatment CBCT. The Kaplan–Meier method estimated local progression-free survival (LPFS), intracranial progression-free survival (IPFS), and overall survival (OS). Log-rank and Wilcoxon–Mann–Whitney tests evaluated inter-group differences, whereas Cox regression analysis assessed prognostic factors. **Results:** Fifty females and fifty males, with a median age of 69 years, received 107 SRTs. A total of 213 BM (range, 1–10 per treatment) with a median volume of 0.22 cc were irradiated with a median minimum BED of 59.5 Gy. The median delivered GTV D95 reduction was −0.3%. The median follow-up was 11 months. Nineteen LP events and a 1-year LC rate of 90.1% were observed. The GTV coverage did not correlate with LC, while the GTV volume was a risk factor for LP, with the 1-year rate dropping to 73% for volumes ≥ 0.88 cc. The median LPFS, IPFS, and OS were 6, 5, and 7 months, respectively. Multivariate analysis showed that patients with melanoma histology and those receiving a second or subsequent systemic therapy line had the worst outcomes, whereas patients with adenocarcinoma histology and mutations showed better results. **Conclusions:** The accuracy and efficacy of the Linac-based SRT approach for BM were confirmed, but the dose distribution alone failed to predict the treatment response, suggesting that other factors must be considered to maximize SRT outcomes.

## 1. Introduction

The incidence of brain metastases (BM) is increasing due to improvements in systemic therapies (STs), which better control extracranial tumor sites and, thus, prolong overall survival (OS) after cancer diagnosis. The brain is the area less responsive to STs because many old-generation drugs do not cross the blood–brain barrier [1] and even novel targeted and immune-modifying agents with better penetration into central nervous system (CNS) tissues have shown increased efficacy when used in combination with local therapies [2,3,4,5,6]. In this context, managing BM with radiation therapy has become extremely important, and recent technologies have enabled replacing whole-brain radiotherapy (WBRT) with localized stereotactic radiosurgery (SRS) as both a primary or combined treatment of up to 10 BM [7,8,9,10], reducing neurotoxicity and improving local control (LC) [11]. Promising one-year LC rates between 69 and 95% have been reported [12], despite a lack of uniformity in terms of therapeutic practices, total dose, number of fractions, target volume, treatment delivery, immobilization, image guidance, planning target volume (PTV) margins, and prescription isodose lines. Higher LC rates (>80–90%) were associated with higher doses and smaller lesions [13], indicating a radiobiological advantage of fractionated SRS (fSRS) for larger lesions [14], in which single doses would exceed those recommended by the Radiation Therapy Oncology Group 90-05 [15]. On the other hand, intracranial progression (IP) and OS in patients who underwent SRS remain challenging, with one-year rates barely surpassing 50–60% [12,13,16]. Patient selection plays a crucial role in enhancing treatment efficacy. EANO-ESMO guidelines [10] identified controlled extra-CNS disease and good performance status (PS) as favorable prognostic factors for recommending SRS treatments. Many other studies [17,18,19,20] determined multiple prognostic indices, including age, number and volume of metastases, and primary tumor histology, to improve the therapeutic decision-making process. However, evidence supporting the identification of which patients will benefit most from SRS remains limited. Based on our prior dosimetric and clinical analysis of a heterogeneous patient population treated with Linac-based single-isocenter stereotactic radiotherapy (SRT), which included both SRS and fSRS [21], this retrospective single-institutional study primarily aims to identify potential prognostic factors associated with treatment outcomes to better inform the decision-making process from both clinical and dosimetric perspectives. Additionally, with an expanded patient cohort, we seek to further assess the efficacy of our treatment strategy. 

## 2. Materials and Methods

### 2.1. Patient Population

This study retrospectively included consecutive patients diagnosed with BM from various primary tumors who underwent SRT in 1 or 3 fractions between March 2020 and December 2023 at a single institution. Patients with lesions too large to meet brain dose constraints with 1- or 3-fraction regimens, subsequently treated with 25–35 Gy in 5 fractions, were excluded from this analysis. Additionally, individuals with a history of previous neurosurgery, as well as those who died within one month after treatment, were also excluded. Patients who received multiple SRT courses for distinct lesions at different times were included, while re-irradiation on the same lesion was not investigated in this analysis.

### 2.2. Simulation, Planning, and Treatment

The SRT characteristics were mostly consistent with those reported in our previous paper [21]. Gross tumor volume (GTV) was determined on post-contrast T1-weighted volumetric magnetic resonance imaging (MRI) fused with planning computed tomography (CT). Doses of 14–21 Gy in 1 fraction or 27 Gy in 3 fractions were prescribed according to current guidelines [10,12] and brain constraints were never exceeded [22]. Treatments were delivered on a VersaHD linear accelerator (Elekta AB, Stockholm, Sweden) employing a single-isocenter volumetric modulated arc therapy (VMAT) technique with 6 MV flattening-filter-free (FFF) coplanar arcs. The institutional image-guided radiotherapy protocol involved pre- and post-treatment cone beam CT (CBCT) scans, with a mid-treatment CBCT conducted for treatments lasting longer than 5 min. Corticosteroids were typically prescribed to prevent edema starting 1 day before treatment and tapered in 3 to 5 days. In the initial patient group, a 2 mm expansion was used to create the PTV, and 2–4 coplanar arcs were applied for treatment planning, ensuring that 99% of the PTV was covered by at least 80% of the prescribed dose with a dose gradient up to 110% within the GTV. Then, following our prior analysis [21], the PTV expansion was reduced to 1 mm and all plans were optimized using only 1 arc, ensuring that 99% of the PTV was covered by at least 90% of the prescribed dose with a dose gradient up to 115% within the GTV.

### 2.3. Clinical and Dosimetric Outcomes

After SRT completion, all patients underwent MRI scans performed at least every 3 months, to follow up the intracranial disease status. Response assessment was based on RANO-BM criteria [23]. Additionally, any instances of systemic progression or shift to the next oncological treatment line were updated. Dose-volume histogram statistics were extracted from the original and delivered plans for each lesion. Delivered doses were estimated by recalculating the original plans on roto-translated CT according to intrafractional errors recorded by post-treatment CBCT, following the method detailed in the aforementioned study [21]. The differences between delivered and planned doses (∆D) estimated the loss of GTV coverage due to delivery inaccuracies. To address the heterogeneity in prescriptions, all GTV doses were converted into biological effective dose (BED) using an α/β value of 10.

### 2.4. Statistical Analysis

The Kaplan–Meier method was used to estimate local progression-free survival (LPFS), intracranial progression-free survival (IPFS), and OS. Progression-free survival was calculated from the date of SRT to the last follow-up or the date of progression. Recurrent disease in the SRT site was considered as local progression (LP), whereas IP included both an LP and/or the appearance of a new distant BM. OS was computed starting from the date of SRT until the last follow-up or death. Log-rank (for categorical variables) and Wilcoxon–Mann–Whitney (for continuous variables) tests evaluated inter-group differences, whereas univariate (UVA) and multivariate (MVA) Cox regression analyses assessed the prognostic role of dosimetric and clinical factors. Arbitrary cut-offs (based on the literature) or interquartile ranges were used to categorize continuous variables and perform inter-group analyses, when deemed appropriate. To determine the feasibility of subgroup analysis between patients treated with the initial and the evolved treatment technique, two-sample *t*-tests (for continuous variables) and chi-square tests (for categorical variables) assessed the balance of covariates. A *p*-value ≤0.05 determined the statically significance level, but all variables with a *p*-value < 0.1 were considered for inclusion in the MVA. Data were analyzed using the software Stata, version 13.0 (StataCorp LLC, College Station, TX, USA).

## 3. Results

### 3.1. Population Characteristics

As a result of the screening stage, a total of 100 patients (50 females and 50 males) remained in the analysis. The median age at the time of BM diagnosis was 69 years (range, 29–86). Patient and tumor characteristics are detailed in Table 1. Since seven patients underwent a second course of SRT, 107 treatment sessions were administered. These sessions comprised 88 SRS and 19 fSRS, amounting to 145 treatment fractions. The prescribed dose had a median of 21 Gy (range, 14–27). A total of 213 lesions were targeted, with a median GTV and PTV volume of 0.22 cc (range, 0.01–8.77) and 0.92 cc (range, 0.12–11.94), respectively. Most lesions (77.9%) were <1 cc in volume, with only 27 BM, 17 BM, and 8 BM with a volume larger than 2 cc, 3 cc, and 4 cc, respectively. The median number of BM per treatment was 1 (range, 1–10), with 54 treatments delivered for a single BM. The remaining 53 treatments involved multiple BM, with the following distribution: 30 with 2 BM, 12 with 3 BM, 3 with 4 BM, 4 with 5 BM, 3 with 7 BM, and 1 with 10 BM. The lesions were predominantly located in the parietal region (65 lesions), with 4 in the brainstem/vermis. The median time from diagnostic MRI to SRT was 25 days (range, 3–97), with 78.7% of the treatments delivered ≥14 days after MRI. Median planned GTV doses are reported in Table 2. Overall, the whole population involved 63 treatments and 117 lesions from the previous cohort and 44 treatments and 96 lesions from the newer cohort, respectively. Statistical tests revealed that the two cohorts were unbalanced in terms of total GTV volume (*t*-test, *p* = 0.012), ST regimen (chi2, *p* = 0.001), number of systemic lines (chi2, *p* = 0.021), and clinical follow-up (*t*-test, *p* = 0.001). Consequently, no subgroup analysis was performed. 

### 3.2. Treatment Outcomes

Median follow-up was 11 months (range, 2–54), during which 43 cases of extracranial progression were documented. After SRT treatment, the ST regimen was changed for 42 patients, with a median time to the next therapy of 3 months (range, 0–46). Nineteen (8.9%) local failure events and a 1-year LC rate of 90.1% (95% confidence interval (CI 95%): 82.9%–94.4%) were observed (Figure 1a). On treatment basis, 17 (15.9%) LP in at least one treated BM and 52 (48.6%) IP occurred. The median LPFS, IPFS, and OS were 6 months (range, 1–47), 5 months (range, 1–46), and 7 months (range, 1–47), respectively. One-year rates were 81.4% (CI 95%: 68.2–89.5%), 41.9% (CI 95%: 29.5–53.8%), and 47.6% (CI 95%: 37.3–57.3%), respectively (Figure 1b–d). At the time of analysis, 39 patients were still alive. The mean GTV dose reductions resulting from intrafractional errors are reported in Table 2, along with the median delivered doses.

### 3.3. Prognostic Factors

#### 3.3.1. LPFS Analysis per Single Lesion

Figure 2 presents all parameters tested in the UVA for LP. One-year rates, divided per D95 interquartile ranges, were as follows: 88.8% for Q1 (23.9–52.7 Gy), 89.1% for Q2 (52.7–62.5 Gy), 89.5% for Q3 (62.6–67.4 Gy), and 90.6% for Q4 (67.4–73.1 Gy). GTV volume (hazard ratio (HR), 1.47, CI 95%: 1.004–1.958; *p* = 0.009) and Dmax (HR, 1.09, CI 95%: 1.018–1.171; *p* = 0.014) remained significant risk factors in the MVA. The median volume was 0.18 cc for lesions that did not recur and 0.91 cc for lesions that recurred (K-wallis, *p* = 0.002). One-year rates, divided per volume interquartile ranges, were as follows: 100% for Q1 (0.01–0.08 cc), 95.8% for Q2 (0.08–0.21 cc), 90.6% for Q3 (0.23–0.87 cc), and 73.0% for Q4 (0.88–8.77 cc). The median Dmax was 69.8 Gy for lesions that did not recur and 70.5 Gy for lesions that recurred (K-wallis, *p* = 0.420). One-year rates, divided per Dmax interquartile ranges, were as follows: 90.5% for Q1 (37.7–59.1 Gy), 91.1% for Q2 (59.3–69.8 Gy), 84.5% for Q3 (70.0–73.8 Gy), and 86.3% for Q4 (73.8–80.8 Gy).

#### 3.3.2. LPFS Analysis per Treatment

UVA revealed that patients with adenocarcinoma (HR, 0.17, 95% CI: 0.059–0.510; *p* = 0.010) and mutated tumors (HR, 0.27, CI 95%: 0.089–0.771; *p* = 0.015) had a better treatment response, while patients with melanoma tumor type (HR, 3.70, CI 95%: 1.313–10.472; *p* = 0.013) exhibited an inferior LPFS. One-year rates were 87.7% vs. 66.4% vs. 72.9% for adenocarcinoma/melanoma/other tumors, and 65.4% vs. 91.9% for wild-type/mutated tumors. Male compared to female patients showed an inferior LPFS, with a 1-year rate of 68.6% vs. 91.9% (log-rank, *p* = 0.045). Of note, 14 male vs. 6 female patients presented melanoma tumors, while 45 female vs. 26 male patients had adenocarcinoma tumors. None of the previous factors were independently related to a longer LPFS in the MVA.

#### 3.3.3. IPFS Analysis per Treatment

Figure 3 presents all parameters tested in the UVA for IP. In the MVA, the presence of mutations (HR, 0.49, CI 95%: 0.246–0.970; *p* = 0.041), systemic line number (HR, 1.70, CI 95%: 1.071–2.713; *p* = 0.024), and ST change post-SRT (HR, 2.10, CI 95%: 1.116–3.949; *p* = 0.021) were independent factors significantly related to IPFS. One-year rates were 24.7% vs. 55.3% for wild-type/mutated tumors, 9.7% vs. 51.0% vs. 54.5% for patients receiving second or higher ST line/first ST line/no ST, and 32.7% vs. 50.2% for patients who changed/did not change ST post-SRT. Multiple BM led to an inferior 1-year IPFS rate compared to a single BM (31.5% vs. 52.1%; log-rank, *p* = 0.031). The median age among patients who experienced/did not experience IP was 63 vs. 74 years (K-wallis, *p* < 0.001). Patients with polymetastatic progression exhibited a trend towards a significantly inferior IPFS (1-year rate: 35.6% vs. 45.6%; log-rank, *p* = 0.064).

#### 3.3.4. OS Analysis per Treatment

Figure 3 shows all parameters tested in the UVA for OS. In the MVA, primary origin tumor (HR, 1.40, CI 95%: 1.041–1.879, *p* = 0.026), systemic line number (HR, 1.59, CI 95%: 1.065–2.370, *p* = 0.023), and ST change post-SRT (HR, 0.59, CI 95%: 0.359–0.995, *p* = 0.048) remained independent factors associated with OS. One-year rates were 57.4% vs. 31.4% vs. 35.6% vs. 42.5% for lung/melanoma/breast/other tumors, 21.3% vs. 41.8% vs. 72.5% for patients receiving second or higher ST line/first ST line/no ST, and 59.0% vs. 38.6% for patients who changed/did not change ST post-SRT. Patients with polymetastatic progression exhibited an inferior 1-year OS rate compared to patients without (37.9% vs. 54.5%; log-rank, *p* = 0.006). Patients with PD-L1 mutations had a 1-year survival rate of 57.7%, whereas those with BRAF mutations had a rate of 22.2%.

## 4. Discussion

In this retrospective single-institutional study, we analyzed dosimetric and clinical prognostic factors in 100 patients with 213 BM treated using 107 courses of SRT, with the aim of improving the decision-making process by potentially determining the optimal dose distribution and identifying patients who derive the maximum benefit from SRT. 

Similar to Nicosia et al. [24], we did not observe a statistically significant correlation between target coverage and LC. Conversely, several retrospective studies [25,26,27,28] have highlighted a close relationship between LC and dose, with 1-year rates exceeding 80% for doses ≥ 21 Gy, over 60% for doses ≥ 18 Gy, and dropping below 50% for doses ≤ 15 Gy. Berthet et al. [29] concluded that BED10 = 50 Gy was a significant threshold for improving the LC rate from 76.5% to 91.6% at 1 year. Similarly, in the Alongi et al. [30] series, BED10 > 51.3 Gy correlated with a higher LC. Our analysis considered the delivered BED GTV coverage, but a higher BED did not appear to be significantly more effective than a lower BED, with 1-year LC rates ranging from 88.8% to 90.6% for the interquartile ranges of D95. The absence of recurrence in the BM treated with lower doses (14–15 Gy) might have influenced this finding. However, considering that most lesions (80.8%) received a BED Dmin >50 Gy, with a median delivered value of 59.5 Gy, it can be inferred that the administered dose was adequate to achieve satisfactory LC. The extent of the overdose within the target was not associated with LC, as no correlations with GTV V107% and V110% were observed, indicating that significant adjustments to the dose distribution might turn out to be helpless. Employing a steep dose distribution increases the risk of target missing due to intrafractional errors, but this did not appear to affect our treatments. The median delivered GTV D95 reduction was −0.3% (range, −14.8%–2.2%) and the analysis of the differences between the planned and delivered doses (∆D) did not find a significant correlation between LC and loss of GTV coverage (Figure 2), thus reassuring the accuracy of the treatment’s delivery.

Like other studies [4,30], our MVA revealed that the BM volume served as a negative prognostic factor for LP. The tumor control probability analysis by Redmond et al. [12] noted decreased rates for medium- and large-sized tumors (>2 cm), suggesting an advantage in multifractionated SRS. Nevertheless, a recent study [31] observed no statistically significant differences in the 1-year LC rate between medium-sized BM (range, 4–14 cc in volume) treated with single-fraction or multifractionated SRS. Moreover, Leyrat et al. [25], analyzing 101 BM (10–46 mm in diameter) treated with the 3-fraction regimen, reported the lower maximum diameter as an independent predictive factor for better LC (HR, 1.15, 95% CI: 1.055–1.259, *p* = 0.002). In this series, although only a small proportion of BM could be classified as medium-sized according to the division by Reinhardt et al. [31], due to the exclusion of larger BM treated in five fractions, the volume trend remained evident. Specifically, the 1-year LC rate decreased to 73% for a GTV volume ≥0.88 cc (Q4), and there was a significant difference in volume between recurring and non-recurring lesions (*p* = 0.002). Future prospective studies should focus exclusively on large lesions to better understand which SRS technique (such as fractionation, staged treatments, dose escalation, or post-surgery interventions) may improve outcomes in these riskier cases. Unexpectedly, Dmax was identified as a risk factor for LPFS, which seems to contrast with the conventional SRS practice of overdosing the tumor center to achieve maximum target control [11,32]. Since larger lesions may require a higher maximum point dose to ensure comparable minimum coverage to smaller lesions in the context of inhomogeneous dose optimization, we hypothesized that Dmax could potentially align with this trend. Even if it emerged as an independent factor in the MVA, inter-group evaluation did not identify statistically significant differences in Dmax between BM that recurred and those that did not, prompting further investigation into alternative contributing factors.

It is worth noting that mutations, tumor histology, and gender were significantly correlated with LPFS at the treatment level, suggesting that dose distribution alone may fail to predict the treatment response, and these characteristics should be considered to maximize SRT outcomes. Different studies [4,30,33] demonstrated that BM from more radioresistant histologies—like melanoma—had the worst LPFS compared to those from lung and breast cancer. In this population, the larger proportion of melanoma among male patients and adenocarcinoma among female patients might also explain the statistically significant gender-based difference in LPFS.

Some studies [34,35] demonstrated a substantial growth rate of BM, and changes in edema over time may significantly alter brain structures and BM locations. Accordingly, an international guideline [11] recommends minimizing the time between planning imaging and SRT delivery. Seymour et al. [34] reported that when MRI images used for contouring were obtained <14 and ≥14 days from SRT delivery, there was a 1-year LPFS of 75% vs. 34% (*p* = 0.0003), respectively. In this study, despite most treatments being delivered ≥14 days after MRI, primarily due to delays in patients’ oncological pathways and/or in scheduling the MRI appointments, Cox analysis did not highlight significant differences in LC depending on time from MRI acquisition to SRT (Figure 2).

Finally, critical factors that were not addressed in the current study included fusion inaccuracies and contouring variations. Although some studies [36,37] showed that the largest differences in target delineation were observed for smaller and multiple metastatic target volumes, the concept of 3D PTV margins may imply that inaccuracies occurring during image fusion could have a greater impact on larger lesions compared to smaller ones. Indeed, the latter may still be adequately encompassed within the PTV, while the former might be more susceptible to extending beyond the PTV margins, thereby increasing the risk of marginal misses. Future investigations could benefit from examining the fusion accuracy and contouring delineation to discern any potential correlations with local recurrences.

Primary tumor histology, genetic mutations, systemic line number, and ST change after SRT were found to be significantly associated with IPFS and OS in our MVA. Moreover, patients with polymetastatic disease had a significantly inferior OS and a trend towards inferior IPFS compared to patients with less advanced and widespread cancer (Figure 3). This finding emphasizes that interventions for BM may offer limited advantages in terms of preventing IP and improving OS, because these outcomes are often influenced by extracranial factors [12,24,30,38]. Barillaro et al. [38], in a population of 87 patients with 220 BM, found that extracranial disease status was the only factor independently related to OS in the MVA (HR, 1.80; CI 95%: 1.020–3.140, *p* = 0.043). Likewise, in another study [33] involving 172 patients with 1079 BM, uncontrolled systemic disease (*p* = 0.000) and melanoma histology (*p* = 0.026) were identified as independent poor prognostic factors for IPFS.

Interestingly, we found that synchronous BM were not related to outcomes, suggesting that patients with simultaneous occurrence of BM alongside the primary cancer did not have reduced chances of survival compared to patients diagnosed with a more limited intracranial disease. Furthermore, immunotherapy, as well as chemotherapy, was identified as a risk factor in the UVA (Figure 3). In this population, longer OS was observed in patients who did not require ST, as the outcomes decreased upon receiving any type of ST regimen. In contrast with other studies [39,40], PS, age, and number of BM did not provide sufficient evidence in this population. The presence of only one instance of PS ECOG 3 may have hindered the identification of significant correlations. Age and number of BM were found to be significantly correlated solely with IPFS, although older patients exhibited significantly longer IPFS compared to younger patients (Figure 3), which contradicts the usual trend reported in the literature [33]. Refining the patient’s selection strategy and/or understanding ST benefits posed challenges, highlighting the need for prospective studies and further research.

Our comprehensive analysis may provide valuable insights for clinicians caring for these patients and it allowed us to thoroughly evaluate the efficacy of our single-isocenter Linac-based method with 6FFF coplanar arcs. Specifically, considering all lesions, only 19 (8.9%) local failure events were observed during the median follow-up of 11 months (range, 2–64). The 1-year LC rate of 90.1% was optimal and consistent with findings from the recent literature [12,13,24,25,29,38,41,42], confirming the effectiveness of our treatment strategy in controlling BM.

This study is not devoid of limitations: firstly, its retrospective nature and the resulting risk of selection bias; secondly, the lack of reporting on the grade of toxicity, including the incidence and severity of radionecrosis or other side effects; and thirdly, the wide range of different histologies and tumor molecular characteristics.

## 5. Conclusions

The accuracy and efficacy of Linac-based SRT for BM with a single-isocenter coplanar FFF-VMAT approach were confirmed. The administered dose distribution proved to be adequate to achieve optimal LC results, with a larger target volume identified as a negative prognostic factor for recurrences. Patients with melanoma histology and those receiving a second or subsequent ST line had the worst outcomes, whereas patients with adenocarcinoma histology and mutations showed better results. These findings suggest that the dose distribution alone may fail to predict the treatment response and other factors must be considered to maximize SRT outcomes.

## Figures and Tables

**Figure 1 cancers-16-03243-f001:**
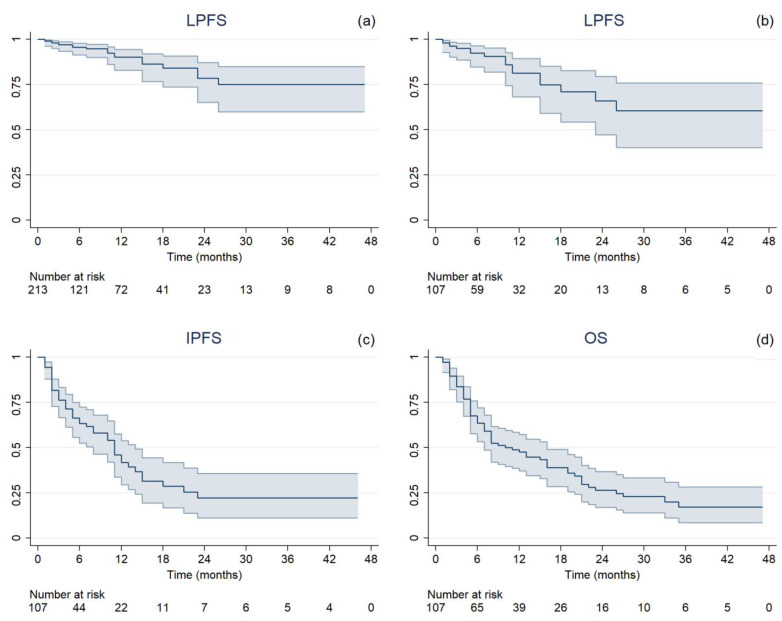
Kaplan–Meier estimated time-to-event curves with 95% confidence interval of (**a**) local progression-free survival per single lesion (LPFS), (**b**) local progression-free survival per treatment (LPFS), (**c**) intracranial progression-free survival (IPFS), and (**d**) overall survival (OS).

**Figure 2 cancers-16-03243-f002:**
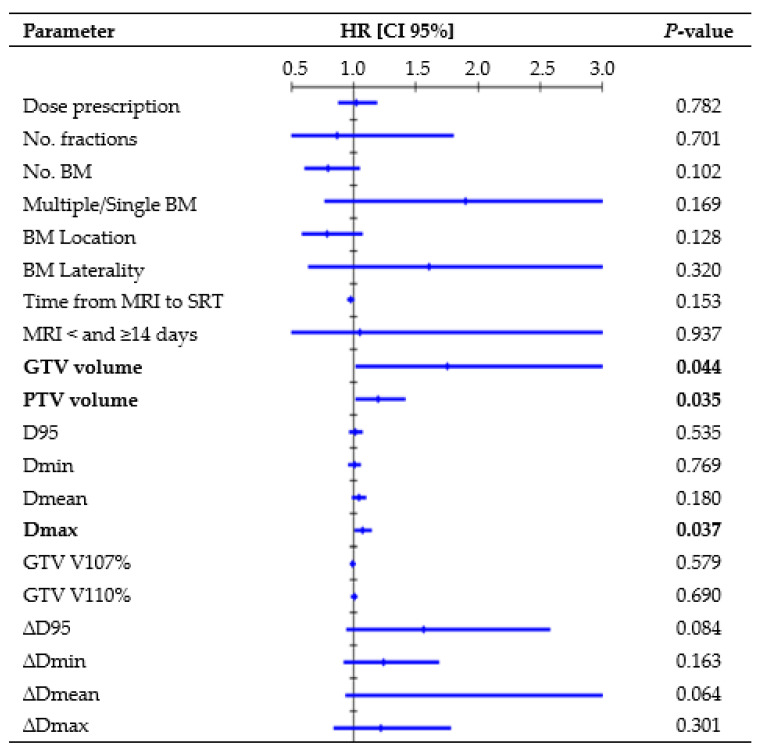
Cox regression univariate analysis of all factors possibly related to local recurrence (performed per single lesion), with statistically significant *p*-values (<0.050) in bold type. Abbreviations: HR = hazard ratio, CI 95% = 95% confidence interval, D95 = dose received by 95% of the volume, Dmin = dose received by 99% of the volume, Dmean = mean dose, Dmax = dose received by 0.035 cc, V107% = percentage of volume receiving 107% of prescribed dose, and V110% = percentage of volume receiving 107% of prescribed dose. All doses were converted to BED10.

**Figure 3 cancers-16-03243-f003:**
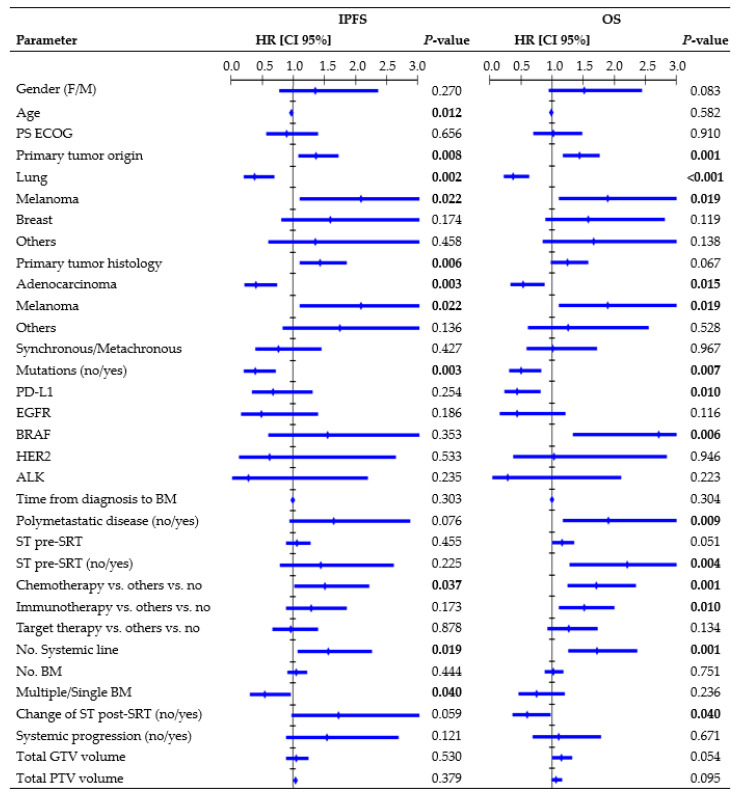
Cox regression univariate analysis of all factors possibly related to intracranial progression and overall survival (performed per treatment), with statistically significant *p*-values (<0.050) in bold type. Abbreviations: HR = hazard ratio, CI 95% = 95% confidence interval, PS ECOG = eastern cooperative oncology group performance status, ST = systemic therapy, SRT = stereotactic radiation treatment.

**Table 1 cancers-16-03243-t001:** Patient and tumor characteristics of the entire patient cohort.

Parameter		No.
Number of patients		100
Primary origin tumor	Lung	53
	Melanoma	18
	Breast	17
	Other	12
Histological type	Adenocarcinoma	66
	Melanoma	18
	Squamous Cell	7
	Other	9
Mutations *	Wild-Type	49
	PD-L1	29 (56.9%)
	EGFR	9 (17.6%)
	BRAF	8 (15.7%)
	HER2	6 (11.8%)
	ALK	3 (5.9%)
PS ECOG	0	33
	1	52
	2	12
	3	1
Median time from the diagnosis of the primary tumor to BM [range]		23 months [0–312]
Timing of BM	Synchronous	69
	Metachronous	31
Polymetastatic progression	Yes	40
	No	60
Pre-RT ST	Yes	70
	No	30
Pre-RT ST regimen **	Chemotherapy	25 (35.7%)
	Immunotherapy	26 (37.1%)
	Targeted therapy	15 (21.4%)
	Hormonal therapy	4 (5.7%)
No. of systemic line **	First-line	43 (61.4%)
	Second-line	18 (25.7%)
	Third-line or higher	9 (12.9%)

* Mutations were identified in 51 patients, with some patients having more than one category of mutation type. The percentages were reported based on this subset of 51 patients, rather than the entire 100-patient population. ** Pre-RT ST was prescribed in 70 patients. The percentages of the pre-RT ST regimen and systemic line number were reported based on this subset of 70 patients, rather than the entire 100-patient population. Abbreviations: PS ECOG = eastern cooperative oncology group performance status, ST = systemic therapy.

**Table 2 cancers-16-03243-t002:** Median GTV coverage and range from original and delivered plans, along with the differences between the two, expressed in both dose and BED_10_.

Parameter	Planned Dose	Delivered Dose	Difference	Planned BED_10_	Delivered BED_10_	Difference
GTV D95%	21.2 Gy	20.9 Gy	−0.3%	62.5 Gy	61.3 Gy	−0.5%
	[11.1–28.9]	[11.2–28.6]	[−14.8–2.2]	[23.9–73.1]	[23.8–72.7]	[−22.9–3.6]
GTV D99%	20.9 Gy	20.6 Gy	−0.4%	61.0 Gy	59.5 Gy	−0.6%
	[9.8–28.8.]	[9.8–28.4]	[−14.9–4.5]	[19.4–73.1]	[19.5–72.7]	[−22.9–1.4]
GTV Dmean	21.8 Gy	21.7 Gy	−0.1%	66.9 Gy	66.9 Gy	−0.1%
	[14.8–30]	[14.7–29.4]	[−6.0–0.9]	[36.6–76.4]	[36.4–76]	[−9.9–1.4]
GTV D0.035 cc	22.5 Gy	22.3 Gy	0.0%	69.8 Gy	69.6 Gy	−0.1%
	[15.0–30.9]	[15.0–30.9]	[−6.7–5.2]	[37.7–80.8]	[37.6–81.7]	[−11.0–−0.3]
GTV V107%	15.4%	8.7%	–	–	–	–
	[0.0–100]	[0.0–94.7]				
GTV V110%	0.0%	0.0%	–	–	–	–
	[0.0–82.9]	[0.0–62.5]				

## Data Availability

Research data are stored in an institutional repository and can be shared upon reasonable request to the corresponding author.

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
