# Peer review of "Dosimetric and Clinical Prognostic Factors in Single-Isocenter Linac-Based Stereotactic Radiotherapy for Brain Metastases"

_cancers, 2024, doi:10.3390/cancers16183243_

Round 1
Reviewer 1 Report
Comments and Suggestions for Authors
You report the results of single isocenter linear accelerator treatment as SRT for metastatic brain tumors in 100 cases at your institution, and I think it would be clinically meaningful to publish the results if the following points are clearly addressed.
In the method, are there any cases that were excluded at the screening stage, 50 male cases and 50 female cases in consecutive cases? Please show the flow from screening to the 100 cases to be considered.
In the polulation characteristic, it says “A total of 213 lesions (range, 1 - 10 per treatment)”, but I think it is necessary to provide more detailed information on the number of lesions for future interpretation of the results of this case. We believe that more detailed information on the number of lesions is necessary for future interpretation of the results of this case. We believe that at least a median value should be given and possibly the number of cases per number of metastases out of 100 cases.
Since most of the cases in this study have a low total GTV, I think the discussion should mention the cases with a high total GTV to increase the clinical significance of this paper.
The Discussion should include a subtitle for each topic to make it easier to understand.
Comments on the Quality of English Language
None
Author Response
Response to Reviewer 1 Comments
You report the results of single isocenter linear accelerator treatment as SRT for metastatic brain tumors in 100 cases at your institution, and I think it would be clinically meaningful to publish the results if the following points are clearly addressed.
Author: Thank you for the time and the efforts you have spent in reading and reviewing our manuscript, and for the positive feedback.
In the method, are there any cases that were excluded at the screening stage, 50 male cases and 50 female cases in consecutive cases? Please show the flow from screening to the 100 cases to be considered.
Author: Our screening stage excluded individuals with a history of previous neurosurgery, as well as those with lesions too large to meet brain dose constraints with 1 or 3 fractions (these cases were subsequently treated with 25-35Gy in 5 fractions), and patients who died within one month after treatment due to the lack of follow-up data. As a result, 50 male and 50 female cases remained in the analysis, without any intentional effort to balance the gender distribution. We have updated the Method and Result sections accordingly.
In the polulation characteristic, it says “A total of 213 lesions (range, 1 - 10 per treatment)”, but I think it is necessary to provide more detailed information on the number of lesions for future interpretation of the results of this case. We believe that more detailed information on the number of lesions is necessary for future interpretation of the results of this case. We believe that at least a median value should be given and possibly the number of cases per number of metastases out of 100 cases.
Author: We agree with your concern and we updated the Result section as follows: “The median number of BM per treatment was 1 (range, 1 – 10), with 54 treatments delivered for a single BM. The remaining 53 treatments involved multiple BM, with the following distribution: 30 with 2 BM, 12 with 3 BM, 3 with 4 BM, 4 with 5 BM, 3 with 7 BM, and 1 with 10 BM.”
Since most of the cases in this study have a low total GTV, I think the discussion should mention the cases with a high total GTV to increase the clinical significance of this paper.
Author: The low GTV volume of this study can be explained by our screening process which excluded patients treated in 5 fractions due to lesions too large to meet brain dose constraints in 1or 3 fractions. Interestingly, even with smaller GTV volumes, the volume trend (larger lesions more likely to recur compared to smaller lesions) remained evident, despite following guidelines that recommend fractionated SRS (3 fractions) for lesions larger than 2 cm. This finding suggest that future studies could benefit from focusing exclusively on large lesions to better understand which SRS technique (fractionation, staged, dose escalation, post-surgery) may improve outcomes in these more challenging and riskier cases. We have added this consideration to the Discussion as per your suggestion.
The Discussion should include a subtitle for each topic to make it easier to understand.
Author: Thank you for pointing this out, we understand the importance of making the text clearer and easier to navigate. However, given the nature of our discussion, where each sentence addresses specific prognostic (clinical or dosimetric) factors along with their analysis and related pitfalls, it becomes challenging to separate these elements into distinct sections with subtitles without disrupting the flow and coherence of the discussion.
Reviewer 2 Report
Comments and Suggestions for Authors
This is a quite interesting paper, presenting the results on the experience with stereotacti radiotherapy in multiple brain metastases. Authors made a good work in presenting the data and analyzed the impact of several factors on prognosis and overall survival. Although some of these factors are still quite elusive, it is clear that accurate planning, timing and evaluation of clinical conditions all make the difference. With one year of overall survival reached, every effort should be done to improve the actual protocols to maximize patients chances for local disease control.
On the basis of what said above, I believe this paper is methodologically valid and should find its way to publication
Author Response
Response to Reviewer 2 Comments
This is a quite interesting paper, presenting the results on the experience with stereotactic radiotherapy in multiple brain metastases. Authors made a good work in presenting the data and analyzed the impact of several factors on prognosis and overall survival. Although some of these factors are still quite elusive, it is clear that accurate planning, timing and evaluation of clinical conditions all make the difference. With one year of overall survival reached, every effort should be done to improve the actual protocols to maximize patients chances for local disease control.
On the basis of what said above, I believe this paper is methodologically valid and should find its way to publication
Author: Thank you for the time and the efforts you have spent in reading and reviewing our manuscript, and for the positive feedback that was really appreciated.
Reviewer 3 Report
Comments and Suggestions for Authors
Dear authors,
This manuscript represents a very interesting approach to statistically study the correlation between the dose and clinical prognostics for brain metastases.
In general, the manuscript finds well written and in a high level of scientific soundness, perfectly translating the main message for this work. The experimental approach is greatly presented and all the details, regarding the patients and the statistic study were provided using a sufficient number of details.
The results and discussion represent a great point in this manuscript. The presentation of this data was extremely careful, and the discussions were provided at high level of details, always using tables and graphs experimentally achieved, besides to updated references to support the points by the authors.
In addition, the conclusions address in a short and direct way that the goal of this study was achieved, pointing out the correlation between the dose distribution and the efficiency of linac-based SRT for BM (brain metastasis).
Furthermore, I suggest the publication of this manuscript, supported in the high-level results achieved, discussion and mainly in the innovation of the authors to propose a study identify potential dosimetry and clinical prognostic factors to better understand the decision-making process in brain metastases.
Thanks loads,
Sincerely,
Author Response
Response to Reviewer 3 Comments
Dear authors,
This manuscript represents a very interesting approach to statistically study the correlation between the dose and clinical prognostics for brain metastases.
In general, the manuscript finds well written and in a high level of scientific soundness, perfectly translating the main message for this work. The experimental approach is greatly presented and all the details, regarding the patients and the statistic study were provided using a sufficient number of details.
The results and discussion represent a great point in this manuscript. The presentation of this data was extremely careful, and the discussions were provided at high level of details, always using tables and graphs experimentally achieved, besides to updated references to support the points by the authors.
In addition, the conclusions address in a short and direct way that the goal of this study was achieved, pointing out the correlation between the dose distribution and the efficiency of linac-based SRT for BM (brain metastasis).
Furthermore, I suggest the publication of this manuscript, supported in the high-level results achieved, discussion and mainly in the innovation of the authors to propose a study identify potential dosimetry and clinical prognostic factors to better understand the decision-making process in brain metastases.
Thanks loads,
Sincerely
Author: Thank you for dedicating your time and effort to accurately read and review our manuscript. We truly appreciate your positive feedback, particularly valuing the detailed and accurate review you provided.
Round 2
Reviewer 1 Report
Comments and Suggestions for Authors
The authors made adequet revision according to my comments, so I think we would accepted this version for publication.